# A survival analysis based volatility and sparsity modeling network for student dropout prediction

Feng Pan[1,2☯], Bingyao Huang[1☯], Chunhong Zhang[1], Xinning Zhu[1], Zhenyu Wu[1], Moyu Zhang[1], Yang Ji[1]*, Zhanfei Ma[2], Zhengchen Li[3]

**1** School of Information and Communication Engineering, Beijing University of Posts and Telecommunications, Beijing, China, **2** School of Information Science and Technology, Baotou Teachers' College, Baotou, Inner Mongolia, China, **3** Department of Personnel, Shenyang Polytechnic College, Shenyang, Liaoning, China

☯ These authors contributed equally to this work.
* jiyang@bupt.edu.cn

**Data Availability Statement:** The data underlying this study, i.e. KDDCup 2015 and XuetangX, come from the publicly available datasets. KDDCup 2015 dataset is available at https://www.biendata.xyz/competition/kddcup2015/data/, and XuetangX

## Abstract

Student Dropout Prediction (SDP) is pivotal in mitigating withdrawals in Massive Open Online Courses. Previous studies generally modeled the SDP problem as a binary classification task, providing a single prediction outcome. Accordingly, some attempts introduce survival analysis methods to achieve continuous and consistent predictions over time. However, the volatility and sparsity of data always weaken the models' performance. Prevailing solutions rely heavily on data pre-processing independent of predictive models, which are labor-intensive and may contaminate authentic data. This paper proposes a Survival Analysis based Volatility and Sparsity Modeling Network (SAVSNet) to address these issues in an end-to-end deep learning framework. Specifically, SAVSNet smooths the volatile time series by convolution network while preserving the original data information using Long-Short Term Memory Network (LSTM). Furthermore, we propose a Time-Missing-Aware LSTM unit to mitigate the impact of data sparsity by integrating informative missingness patterns into the model. A survival analysis loss function is adopted for parameter estimation, and the model outputs monotonically decreasing survival probabilities. In the experiments, we compare the proposed method with state-of-the-art methods in two real-world MOOC datasets, and the experiment results show the effectiveness of our proposed model.

## Introduction

The spread of Covid-19 has brought new opportunities for e-learning, especially for the online education platforms represented by Massive Open Online Courses (MOOCs), which attract people's attention once again. However, due to the scarcity of adequate supervision and necessary restrictions for e-learners, online students have a much higher chance of dropping out than those attending conventional classrooms [1]. To address the issue, researchers have proposed Student Dropout Prediction (SDP), which utilizes students' demographics and engagement data to automatically predict whether a specific student will drop out of the MOOC

dataset is available at http://moocdata.cn/data/user-activity. Both datasets are provided by XuetangX platform (https://www.xuetangx.com/), and have been anonymized to meet the requirement of data privacy and ethical issues. The author Feng Pan and Bingyao Huang have accessed the datasets and preprocessed the original data. The processed data are available in the Zenodo repository: https://doi.org/10.5281/zenodo.5914059. The authors did not have any special access privileges. Interested researchers will be able to access the data in the same manner as the authors. Interested researchers will also be able to replicate the results of this study by following the protocols at dx.doi.org/10.17504/protocols.io.b4duqs6w.

**Funding:** This research is supported in part by Key-Area Research and Development Program of Guangdong Province (Grant No: 2020B0101130013), in part by National Natural Science Foundation of China (Grant No: 61762071), and in part by Baotou Teachers' College High Level Research Incubation Project (Grant No: BSYKJ2021-WY04). The funders had no role in study design, data collection and analysis, decision to publish, or preparation of the manuscript.

**Competing interests:** The authors have declared that no competing interests exist.

platform [2]. The accurate and timely prediction can help institutions, students, and faculty members to find more efficient methodologies to mitigate withdrawals.

Traditional approaches for modeling SDP are generally based on machine learning algorithms such as Logistic Regression(LR) [3] and Support Vector Machines (SVM) [4] to fit time-independent features. Unfortunately, these methods often lose the ability to capture complex temporal dependency between students' interactive activities. Recently deep learning architectures such as Convolutional Neural Network (CNN) [2] and Recurrent Neural Network (RNN) [5] have shown the promising capability of modeling students' dynamic activity sequences (time-varying covariates). While exhibiting performance improvements, most current methods treated SDP as a binary classification task, facing limitations in providing a single prediction outcome. However, student dropout is not a sudden event but a long process affected by time. It can be handled most effectively by survival analysis techniques, which could estimate survival (failure) as a function of time. For example, Ameri et al. [6] applied typical survival analysis techniques, i.e., Cox proportional hazards model (COX) and Time-Dependent COX (TD-COX), to fit pre-enrollment and semester-wise information for classifying which student is going to drop out and estimating when this is going to happen. Wintermute et al. [7] modeled the certificate rates of MOOC users with a Weibull survival function, following the intuition that students "survive" in a course for a particular time before stochastically dropping out. However, literature on SDP tends to assume that survival estimates follow a specific parametric distribution, which may not work in all cases. To solve this issue, the seminal work [8] extended the previous assumptions by using RNN to learn the parametric distribution of both survival time and user activity sequences, which provides us valuable new insights into combining neural networks and survival analysis to solve the SDP problem. Although this approach has more flexibility to handle the survival analysis modeling, it is still challenging due to many aspects that affect the accuracy of estimates, such as data volatility and sparsity.

First, the **volatility** in data may cause heteroskedasticity, which affects the model reliability [9]. In recent years, in some human-related application scenarios such as MOOCs, it has been independently observed that a series of user actions burst together in a short period, making peaks randomly appear in time series [10, 11]. The phenomenon may cause the non-stationary variance (heteroskedasticity, also known as volatility), leading to difficulty in sequential modeling [9]. On that account, volatility modeling has attracted sparked interest in the research community [12, 13]. The most common methodologies found in literatures treated the peak points as outliers [14, 15] and adopted anomaly detections to identify them. Then they could be removed manually or replaced with the means of a moving window. However, it always requires extensively manual work for continuous re-detection. More importantly, the burst in activities may indicate users' underlying intention, which should be taken into account rather than removed or replaced.

Second, the **sparsity** of data makes the RNN based model have difficulty in handling sparse time series [16]. In addition to bursts of rapidly occurring events, human-created time series datasets are always separated by long inactive periods, showing irregular sparsity [17, 18]. It presents a more significant challenge because sparse data and the resulting missing values severely limit the data's ability to be analyzed and modeled for classification and forecasting tasks [19]. The sparsity modeling problem has been widely studied, and existing approaches mainly involve two categories. The first category centered around filling in the missing values with substituted values at the data pre-processing stage, which is known as data imputation [20, 21]. However, these pre-processing techniques contaminate the training data significantly when the individual time series is highly sparse and irregular. Moreover, they have the apparent limitation of disregarding important information between observations and missing values. Accordingly, the second category tried to resolve this obstacle by modifying the gating

architecture of LSTM or GRU units to handle missing values in the learning process directly [22, 23]. However, these works are restricted in addressing specific dataset issues that limit transferability to different application domains.

To the best of our knowledge, there is currently no method that can inherently provide a mechanism to tackle the challenges mentioned above. Therefore, we address the following research questions:

- RQ1: Is it possible to model the temporal dynamics of user activities while addressing the issues of volatility and sparsity without additional feature pre-processing?

- RQ2: Can the treatment of volatility and sparsity issues improve the predictive performance of the state-of-the-art survival analysis models in SDP tasks?

In this paper, we aim to answer these questions by addressing the volatility and sparsity issue of sequential modeling in an end-to-end deep learning framework. In particular, we design a Survival Analysis based Volatility and Sparsity Modelling Network (SAVSNet) to model the SDP task. SAVSNet consists of three stages: the volatility modeling network, the sparsity modeling network, and the survival analysis network. The volatility modeling network adopts a 1D-CNN to filter the original signal's volatility adaptively. Parallelly, an LSTM is applied to capture the temporal dependency of input time series. Then, we employ an update gate to fuse the hidden representation output by these two branches adaptively. This approach uses convolution to filter the volatility and preserves the temporal information extracted by the LSTM. In the sparsity modeling network, we handle the sparsity in a customized LSTM unit called Time-Missing-Aware LSTM (TM-LSTM), which concatenates informative miss-ingness patterns, i.e., missing data indicators and time intervals, with the time series, rather than performing simple imputations. Our model captures the long-term temporal dependency of time series observations and utilizes the missing patterns to improve prediction perfor-mance. With the output of the second stage, we compute the hazard rate for each timestep, which indicates the instantaneous dropout rate of a student at time $t$ given that no event has occurred before. Afterward, a survival analysis loss function is employed to estimate the hazard rate. The corresponding survival probability that indicates no dropout event has occurred by time $t$, is derived from the predicted hazard rate. In comparison with binary classifiers, which generally provide single outcome prediction, the benefit of using a survival analysis framework is that it can provide continuous and consistent prediction results [8].

We conduct experiments on two real-world student dropout datasets to demonstrate the effec-tiveness of our proposed model. The results show that our model outperforms the state-of-the-art models. We also conduct an ablation study to reveal the effectiveness of the model components and a sensitive experiment to explore the effect of hyperparameter settings on the model.

To summarize, our contribution to this work is threefold:

1. We propose an end-to-end learning approach for modeling the volatile time series without the need for an additional data smoothing step. This approach not only utilizes the spatial feature extraction capability of 1D-CNN to perform adaptive filtering on the original data but also retains useful temporal information through LSTM.

2. We propose a novel TM-LSTM unit that effectively exploits two representations of infor-mative missingness patterns in the sparse time series to improve prediction performance. This method avoids a separate interpolation process and can be incorporated into other deep learning frameworks.

3. As far as we know, we are the first to combine neural networks and survival analysis to solve the SDP problem. Our approach takes advantage of the neural network's ability to

capture the potential non-linear relations between the hazard rates and time-varying covariates without any specific survival time distribution assumption.

## Related works

In this section, some classical methods of student dropout prediction are introduced first. Then, we list the survival analysis methods proposed so far. Finally, we review two lines of work relevant to ours, namely volatility and sparsity modeling.

### Student dropout prediction

In the surveyed works that have addressed the SDP problem, most researchers treated SDP as a binary classification task and used classic Machine Learning (ML) algorithms to estimate the classification probability of a given vector of features. For instance, Gitinabard et al. [3] use decision trees to select features and use logistic regression to predict dropouts. Ayouni et al. [4] trained Decision Tree (DT) and Support Vector Machines (SVM) to classify a student into different engagement levels. However, these methods remain a need for meticulously designed feature engineering, limiting their applicability. Moreover, they may fail to model the complex nonlinear relations of dynamic student activities.

As deep learning (DL) models have successfully modeled complex nonlinear dependencies, there has been increasing research effort to apply DL algorithms to SDP. The clickstream information is commonly used by the academia [24], and RNN-based models are often chosen due to their ability to capture the dynamics of sequential patterns. Wang et al. [5] approached the SDP task as a sequence classification problem and used the RNN model to obtain a better prediction result. CNN can also be found in the literature to eliminate the potential inconsistency introduced by manual feature engineering. The work of [2] proposed a CFIN framework, which employed a one-dimensional CNN (1D-CNN) to extract features from an augmented learning activity feature vector automatically.

Despite the success of the methods mentioned above in modeling the binary classification task, they may give rise to a biased estimation when modeling the instances that the dropout event is not observed due to time limitations [25]. It is known as the censoring problem, which survival analysis techniques could effectively handle.

### Survival analysis

Survival analysis (SA) models have been successfully applied in a variety of real-world domains such as biology/medicine analysis, credit risk, student/employee attrition, and predictive maintenance that require estimating the time until an event of interest occurs [25]. One of the main challenges for modeling such time-to-event data is the so-called censoring problem, and SA addresses such problem by learning the dependencies between covariates and survival time.

The most commonly used SA model is the Cox Proportional Hazards (CPH) model [26]. It is a semi-parametric model, which does not require any knowledge of the underlying distribution of time, but assumes that the linear combination of the example covariates have an exponential influence on the outcome. Recently [6, 7], have introduced the CPH model into the SDP domain to predict the risk of students dropping out of their degrees. However, in practice, the linear method may fail to fit the dependencies between covariates and survival time. This problem is solvable to some extent through the use of ML techniques. For instance, Nasejje et al. developed a Conditional Survival Forest (CSF) model, which was an ensemble of tree-based SA model [27]. It tends to recursively partition data based on similar events of interest.

Katzman et al. proposed a DeepSuvr model [28], which utilized a three-layer fully connected neural network to fit the nonlinear relationship between input features and the survival status of a subject.

Although these extensions of the CPH model have partially solved the problem of fitting non-linear relationships, they are still semi-parametric models, and the distribution of survival time is still unknown. Accordingly, parametric methods have been proposed, assuming the survival time of all instances follows a particular distribution. A case in point is the Gompertz distribution [29]. It assumed that the logarithm of the hazard follows a skewed continuous probability distribution, resulting in an exponentially increasing failure rate. However, assuming particular distribution for survival time may not apply to all situations. Accordingly, Zheng et al. proposed a neural SA model for fraud early detection (SAFE) [8]. It utilized RNN to learn the parametric distribution of both time and covariates without any specific survival time distribution assumption. However, the presence of volatile and sparse time series can severely limit the deep sequential model's ability for classification and forecasting tasks.

## Volatility and sparsity modeling

A characteristic of time series volatility is the random presence of peaks that deviate from the average over temporal- or channel-wise. Modeling with the peaks can significantly affect the model performance [15]. Therefore, the most straightforward option in the SDP field was to treat the peak data as outliers and use outlier detection methodologies such as clustering [30] and Exceptional Model Mining (EMM) [31] to identify and remove them. An alternative approach was to use a moving window average to smooth the anomalous peak in time series and reduce short-term noise [32]. However, these methods split the detection and prediction into two separate steps, increasing time and computational complexity. What's more, It is not appropriate to remove or smooth these peaks without considering the reasons for their creation. Actually, the peak data in the MOOC is a "natural outlier", which may be generated by genuine learning activity [15]. It may provide helpful information about students' learning and should be considered when making predictions.

Sparsity is characterized by a large proportion of missing data in a dataset, and the intervals between successive observations are often irregular. It always causes performance degradation for classification and prediction tasks. One way to solve this issue is to deploy data interpolation techniques such as spline [20], and Generative Adversarial Networks (GAN) [21] to get data readings of the same interval. However, these methods tend to perform worse when datasets have a high proportion of missing values. In fact, the missing values in MOOCs are a noteworthy indicator of the dropout trend, which should be integrated into the model rather than replaced by interpolation. Accordingly, some studies have integrated missing pattern information into complex temporal dependency learning processes by modifying conventional gated GRU or LSTM architectures. Che et al. [22] developed a modification of the GRU unit, namely GRU-D, which introduced a missing indicator and time interval to impute missing values as the decay of previous input values toward the empirical mean. Although GRU-D considered two representations of missing patterns, they were only applied to generate a derived input for interpolation. It may seriously compromise the model's reliability as it cannot distinguish imputed values from actual records. Baytas et al. [23] proposed a Time-Awre LSTM (T-LSTM) unit, which employed the elapsed time between consecutive elements of a sequence to decay the short-term memory of the standard LSTM unit rather than the current input. This approach has captured the global structure information of irregular time series. However, it is restricted in transferability to different application domains due to only considering the effect

of the short-term memory on the current output, and the decay mechanism needs to be explicitly designed.

## Methods

### Formulation of student dropout prediction

Unlike previous work that modeled the SDP problem as a binary classification task, our study is particularly interested in developing a model based on the survival analysis framework, which could predict the dropout outcome and output monotonically decreasing survival probabilities to achieve consistent predictions along time. Meanwhile, we aim to perform end-to-end learning directly using volatile and sparse MOOC time series as input without the need for an additional interpolation or data smoothing step.

Below we first provide some of the terms that used in this study.

- **Dropout**: A student who does not have any engagement records after a specific point-in-time is defined as a dropout.

- **Event**: Student dropping out of the course is our event of interest.

- **Duration**: It corresponds to the length of time a user is active, i.e., the time between the first observation of student activities and the end of learning (active withdrawal or the end of the observation window).

- **Censored**: A student who has not been subject to a dropout event in the observation window is considered a censored instance. Therefore, it can be found that the dropout and censored indicators are mutually logical non-s.

Formally, we present the notations used in this paper.

Let $\mathcal{D} = \left\{ (\boldsymbol{X}^i, y^i, T^i) \right\}_{i=1}^N$ represents a dataset with $N$ samples, where $T^i$ denotes the duration time and $y^i$ is the ground truth that indicates the sample $i$ is a dropout ($y^i = 1$) or a non-dropout ($y^i = 0$). For each sample $i$, we denote a multivariate time series with $D$ variables of length $T^i$ as $\boldsymbol{X} = (\boldsymbol{x}_1, \cdots, \boldsymbol{x}_t, \cdots, \boldsymbol{x}_{T^i})^{\mathrm{T}} \in \mathbb{R}^{T^i \times D}$, where for each $t \in \{1, \ldots, T^i\}$, $\boldsymbol{x}_t \in \mathbb{R}^D$ represents the $t$-th observations of all variables and $x_t^d$ denotes the measurement of $d$-th variable of $\boldsymbol{x}_t$. The goal of our model is to train a mapping function between the input $\boldsymbol{x}_t$ and the survival probability at each timestamp, i.e., $S_t = f(\boldsymbol{x}_t)$. By comparing the survival probability $S_t$ at time $T^i$ with a threshold $\tau$, we can obtain a binary outcome $y^i$.

### Model structure

In this section, we will describe the architecture of the SAVSNet model. We first give an overview of SAVSNet. Then we will describe the main components.

**Overall architecture.** Fig 1 presents the architecture of our proposed model SAVSNet, which is designed as three stages of deep learning networks: a volatility modeling network, a sparsity modeling network, and a survival analysis network. (1) In the volatility modeling network, we leverage a dual-branch structure consisting of a 1D convolutional network to capture the spatial dependencies of multivariate time series and an LSTM network to learn the temporal dependency, respectively. Then we utilize an update gate to fuse the two hidden representations adaptively. (2) In the sparsity modeling network, we design a customized Time-Missing-Aware LSTM (TM-LSTM) unit that exploits the elapsed time and missing indicator to adjust the long-term and short-term memory of the LSTM unit. (3) In the survival analysis network, we apply a softplus function to transform the output of TM-LSTM into a hazard rate that represents the instantaneous hazard of the student given no dropout event occurred before. Then

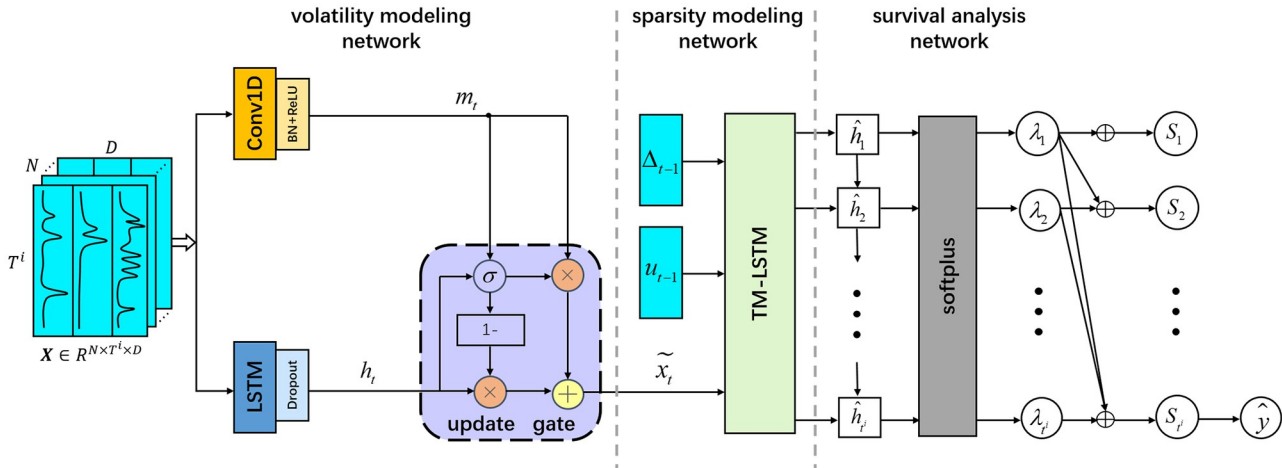

**Fig 1. The architecture of SAVSNet.** The model consists of three major components, namely the volatility modeling network, the sparsity modeling network, and the survival analysis network.

we employ a survival analysis loss function to estimate the hazard rates. Accordingly, the survival probability at each time step is obtained by the formula $S_t = e^{-\sum_{k=1}^{t} \lambda_k}$ [8]. By comparing the survival probability at the last timestamp with a threshold $\tau$, we can predict whether a user would drop out of the course.

**Volatility modeling network.** A common practice in existing approaches for volatility modeling is to treat the local and temporary peaks as outliers and use outlier detection techniques to identify and remove/smooth them, respectively. However, these approaches always lead to a two-step process and cannot be integrated into a deep learning architecture. Therefore, we seek to develop a unified deep learning approach to filter the multivariate time series peaks. Experiment from [33, 34] has shown that CNN is insensitive to data variation through its spatial dependency extraction capability. It inspires us to utilize a 1D-CNN network to capture the spatial dependency of time series to reduce the feature scale variations. In addition, considering the possibility of losing useful information by filtering the peak data, we also use an LSTM network parallel to 1D-CNN to capture the temporal dependency of the original time series. Then, the spatial and temporal dependencies are fed into an update gate to generate a hidden representation of the original data.

The 1D-CNN network receives the MTS $X$ as input and applies filters to convolve with the sequence input. The convolution operation of 1D-CNN is illustrated by Fig 2. A filter produces a 1D feature map by converging over the local area of the input signal, i.e., temporal-wise and channel-wise features. Then, the convolution operation slides the filter across the sequence to detect specific characteristics in all locations. Each filter will constantly adapt to the input signal, effectively matching its filter coefficients to a short-term model of the signal source, thereby reducing the mean square error output. This operation preserves feature scale invariance and can be regarded as a form of adaptive filtering [35]. The 1D-CNN is succeeded by a Batch Normalization (BN) layer to accelerate and stabilize the training process, and the BN layer is followed by a Rectified Linear Unit (ReLU) activation function to ensure the non-linear behavior of the network. The convolution operation is formalized in the equation:

$$m_t = \text{ReLU}(BN(b_t + \sum_{j=1}^{N_l - 1} Conv1D(\omega_{jt}, \boldsymbol{X}))) \tag{1}$$

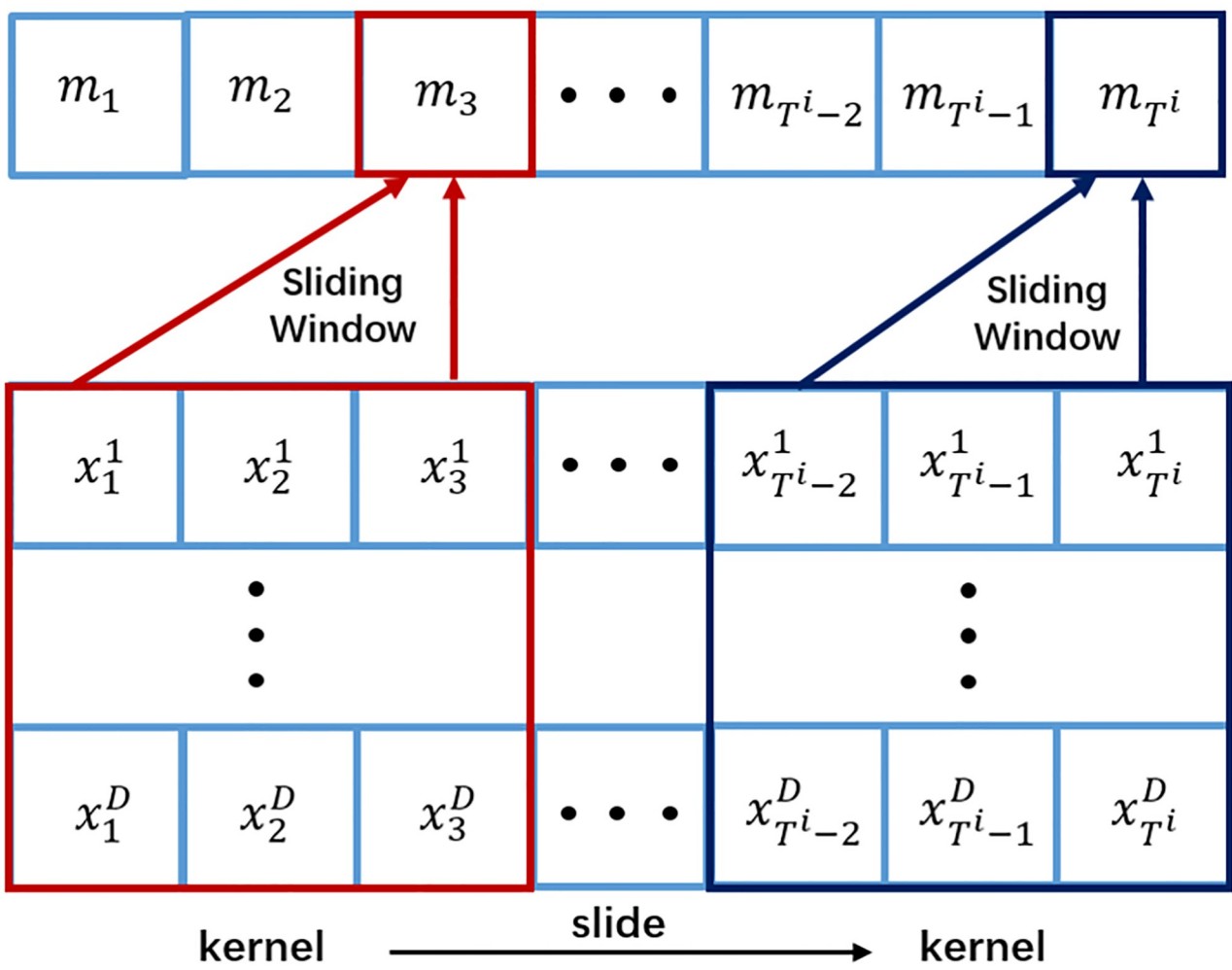

**Fig 2. An example of the convolution operation of 1D-CNN.** 1D-CNN moves the convolution kernel smoothly along the temporal wise of the input time series and generates the filtered signal by performing convergence over the local area of the input.

where $m_t$ is the feature maps of output, $b_t$ is the bias, $N_l$ is the number of channels and $\omega_{jt}$ is the filter weight.

The LSTM network takes the token $\boldsymbol{x}_t$ at each time step as input and outputs the hidden state $h_t$, which captures temporal dependency among the MTS input. A dropout layer succeeds the LSTM to prevent neural networks from overfitting. The computation of LSTM at each time step is depicted by [36] as the following:

$$
\begin{aligned}
f_t &= \sigma_g(W_f \boldsymbol{x}_t + U_f h_{t-1} + b_f) \\
i_t &= \sigma_g(W_i \boldsymbol{x}_t + U_i h_{t-1} + b_i) \\
o_t &= \sigma_g(W_o \boldsymbol{x}_t + U_o h_{t-1} + b_o) \\
\tilde{c}_t &= \sigma_c c(W_c \boldsymbol{x}_t + U_c h_{t-1} + b_c) \\
c_t &= f_t * c_{t-1} + i_t * \tilde{c}_t \\
h_t &= o_t * \sigma_h(c_t)
\end{aligned}
\tag{2}
$$

where $\{W_f, U_f, b_f\}$, $\{W_i, U_i, b_i\}$, $\{W_o, U_o, b_o\}$, and $\{W_c, U_c, b_c\}$ are the network parameters of the forget gate $f_t$, input gate $i_t$, output gate $o_t$ and the candidate memory $\tilde{c}_t$. $c_t$ is the cell state and $h_t$ is the hidden state.

With the output of 1D-CNN and LSTM, we employ an update gate to adaptively fuse them to derive a representation of the original data. The design of the update gate is inspired by the GRU [37], which uses a gating mechanism to control the weights about how much information from two features will be preserved. The process can be formulated as follows:

$$
\begin{aligned}
z_t &= \sigma([m_t, h_t]W_z + b_z) \\
\tilde{x}_t &= z_t \odot m_t + (1 - z_t) \odot h_t
\end{aligned}
\tag{3}
$$

where $[., .]$ represents the concatenating operation and $\odot$ means the Hadamard product.

Since the concatenating operation needs to ensure the two joined elements are consistent, we set the 1D convolutional layer with a stride of 1 and "same" padding, making the sequence length $d_h$ generated by the 1D-CNN equal to that of LSTM. Then, we concatenate $m_t$ with $h_t$ into a vector with length $2d_h$. Then, we send the combination vector into a fully connected layer to generate an update gate vector $z_t$ with output size $d_h$. The vector $z_t$ is then used to control the proportion of $m_t$ and $h_t$ in the output hidden state $\tilde{x}_t$.

**Sparsity modeling network.** Gated RNN variants with different gates and inner connections have become a promising tool for handling the sparsity issue of irregular time series. These variants made effective use of missing value patterns, time intervals, and complex temporal dependencies to achieve improved performance in terms of speed and accuracy for specific datasets or domains [19]. The work of T-LSTM [23] introduced a solution by decaying the short-term memory of the LSTM unit according to the elapsed time intervals between two consecutive records. However, only considering the effect of sparsity on short-term memory may have limited transferability to different application domains. As listed in Eq (2), there are two forms of cell memory in the standard LSTM unit. One is long-term memory, i.e., the cell state $c_t$, which aggregates data from all previous timesteps through linear computation. The other is short-term memory, i.e., the hidden state $h_t$, which encodes the characterization of the previous timestep's data through nonlinear activation. When modeling sparse time series using standard LSTM units, long-term memory would not be significantly affected by the previous record with a more extended period. In contrast, even the input zero vector will cause the LSTM unit to generate non-zero output values due to the nonlinear mapping of short-term memory, which could make data inferred from sparse observations less reliable.

Therefore, we aim to explore a new unified approach to address both of these issues. We introduce two representations of informative missing patterns, i.e., missing indicator and elapsed time to regulate the flow of information in the LSTM unit. The missing indicator $u_t \in \{0, 1\}^D$ is a binaray vector that advises the model which inputs are observed (or missing) at time step $t$. To be specific, if the input token $x_t$ is observed with non-zero statistics, the indicator is set to 1. Conversely, if $x_t$ is filled with zero vectors, the indicator is set to 0.

Thus we have:

$$
u_t =
\begin{cases}
1, & \text{if } x_t \text{ is observed with non} - \text{zero} \\
0, & \text{otherwise}
\end{cases}
\tag{4}
$$

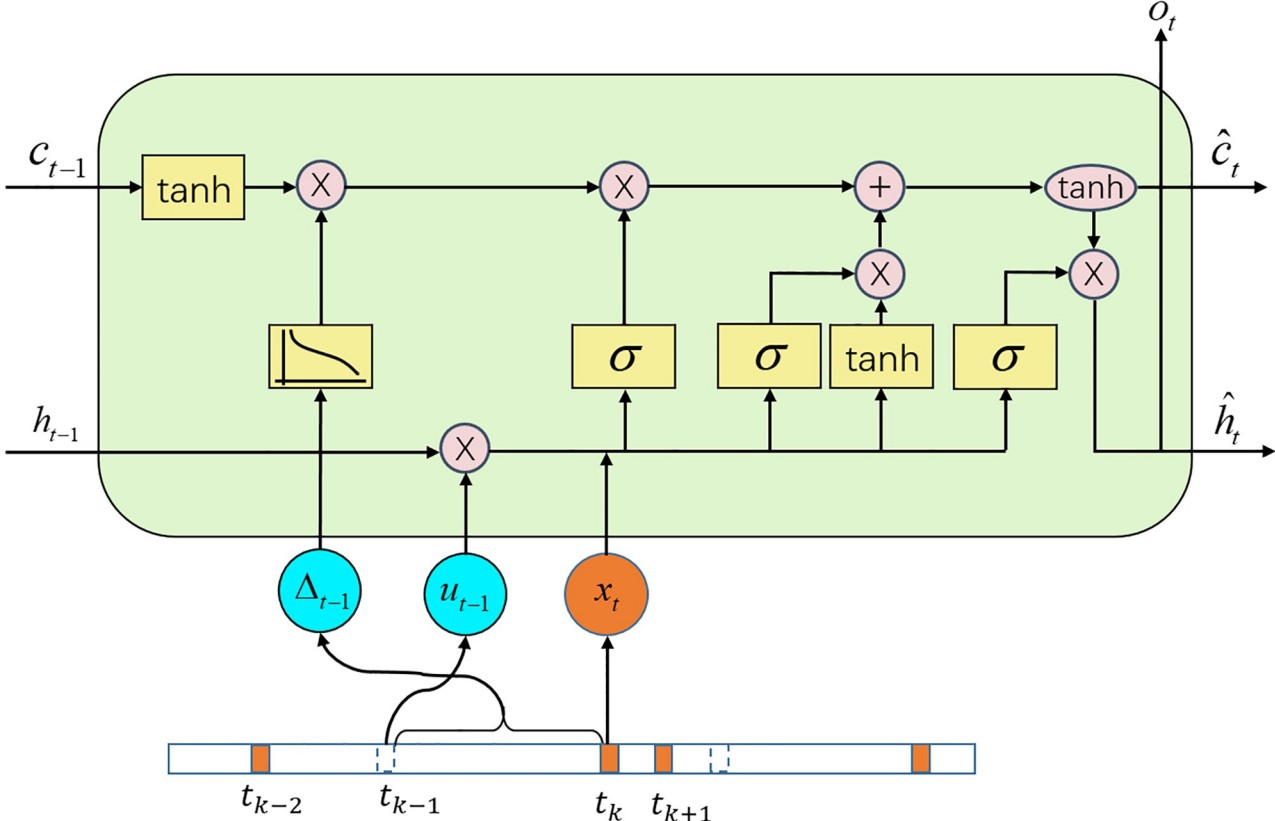

**Fig 3. The structure of Time-Missing-Aware LSTM (TM-LSTM) cell.** TM-LSTM introduces the elapsed time $\Delta_{t-1}$ and the missing indicator $u_{t-1}$ to adjust the previous long-term and short-term memory, respectively.

With the missing indicator $u_t$, we can calculate the elapsed time $\Delta_t \in \mathbb{R}^D$ to stand for the time gap from the last observation to the current timestamp.

$$\Delta_t = \begin{cases} 0, & \text{if } t = 0 \text{ or } u_t = 1 \\ \Delta_{t-1} + 1, & \text{if } u_t = 0 \end{cases} \tag{5}$$

Then, we propose a Time-Missing-Aware LSTM (TM-LSTM) unit to incorporate the two informative missing patterns into the recurrent neural network. The struture of TM-LSTM unit is illustrated by Fig 3. In the comparison with the classical LSTM unit, TM-LSTM takes three inputs, which are the output of the previous stage $\tilde{\mathbf{x}}_t$, the missing indicator $u_{t-1}$ and the elapsed time of previous time step $\Delta_{t-1}$.

TM-LSTM employs $\Delta_{t-1}$ to adjust the confidence of previous long-term memory $c_{t-1}$. Intuitively, longer the elapsed time, less influence of long-term memory has on the current output. So we use a monotonically non-increasing function $g(\cdot)$ to convert the elapsed time to a decaying weight. Here we select $g(\Delta t) = 1/\log(e + \Delta t)$ as suggested in [38]. Meanwhile, we put the previous long-term memory $c_{t-1}$ through a tanh activation to obtain a representation $C_{t-1}^L$ by the network. Consequently, we can obtain a discounted long-term memory $\hat{C}_{t-1}^L$ by the

product of $C_{t-1}^{L}$ and $g(\Delta t)$.

$$C_{t-1}^{L} = \tanh(W_d C_{t-1} + b_d)$$

$$\hat{C}_{t-1}^{L} = C_{t-1}^{L} * g(\Delta_t)$$

(6)

where $\{W_d, b_d\}$ are the network parameters to be learned.

Meanwhile, TM-LSTM applies $u_{t-1}$ to control the flow of previous short-term memory $h_{t-1}$. As previously analyzed, the presence of zero vectors could make the recurrent cells generate non-zero hidden state output. Hence, we use the missing indicator as a switch to control the transfer of short-term memory by a pointwise multiplication. Specifically, $u_{t-1}$ equal to 1 indicates that there are non-zero inputs at the previous time step, so $h_{t-1}$ is the nonlinear mapping result of the previous inputs, and it should be let through to participate in the calculation of the current memory. While $u_t$ equal to 0 means that the input of the previous time step is an all-zero vector, so $h_{t-1}$ is derived entirely from the transmission of the previous hidden state, which could make it less reliable in characterizing the hidden state at the previous step. Therefore, the previous short-term memory $h_{t-1}$ should be masked, and the transmission of memory is entirely dependent on the long-term memory.

$$h_{t-1}^{S} = h_{t-1} * u_{t-1}$$

(7)

The adjusted previous long-term memory $\hat{C}_{t-1}^{L}$ as given in Eq (6) and the adjusted previous short-term memory $h_{t-1}^{S}$ as given in Eq (7) are then injected into a standard gated architecture of the vanilla LSTM:

$$f_t = \sigma(W_f \tilde{\mathbf{x}}_t + U_f h_{t-1}^{S} + b_f)$$

$$i_t = \sigma(W_i \tilde{\mathbf{x}}_t + U_i h_{t-1}^{S} + b_i)$$

$$o_t = \sigma(W_o \tilde{\mathbf{x}}_t + U_o h_{t-1}^{S} + b_o)$$

$$\tilde{C} = \tanh(W_c \tilde{\mathbf{x}}_t + U_c h_{t-1}^{S} + b_c)$$

$$\hat{C}_t = f_t * \hat{C}_{t-1}^{L} + i_t * \tilde{C}$$

$$\hat{h}_t = o_t * \tanh(\hat{C}_t)$$

(8)

where $\hat{C}_t$ and $\hat{h}_t$ is the cell state and hidden state of TM-LSTM, and the gates and network parameters are interpreted in the same way as Eq (2).

**Survival analysis network.** Instead of converting the output of the sparsity modeling network into a binary prediction result, we introduce a survival analysis network, which maps these output hidden representations to survival probabilities to address the issue of censoring and achieve consistent prediction results.

The survival analysis network starts with a softplus layer to transform the hidden state $\hat{h}_t$ at each timestamp to a positive hazard rate $\lambda_t$. This is done because the hazard rate $\lambda_t$ represents the instantaneous probability at time $t$ given no event occurred before time $t$ [8]. The equation is given as below:

$$\lambda_t = \text{softplus}\left(w_\lambda \hat{h}_t\right) = \ln\left(1 + \exp\left(w_\lambda \hat{h}_t\right)\right)$$

(9)

where $w_\lambda$ is the weight of $\hat{h}_t$. Then, we adopt a maximum likelihood function as used in literature [8] to estimate the hazard rate and the corresponding survival probability. If a sample $i$

has the dropout event ($y^i = 1$), the likelihood function seeks to make the predicted time-to-event equal to the accumulation of all hazard rates before $T^i$ as maximizing the term $P\{T < T^i\}$; if a sample $i$ is a non-dropout ($y^i = 0$), the likelihood function aims to make the sample survive over the last-observed time $T^i$, i.e., maximizing $P\{T \geq T^i\}$. The joint likelihood function for a sample $i$ is (A detailed derivation is provided as S1 File):

$$
\begin{aligned}
& P\{T < T^i\}^{y^i} \cdot P\{T \geq T^i\}^{1-y^i} \\
& = (e^{\sum_{t=1}^{T^i} \lambda_t} - 1)^{y^i} \cdot e^{-\sum_{t=1}^{T^i} \lambda_t}
\end{aligned}
\tag{10}
$$

where $T$ is a continuous non-negative random variable representing the survival time until the occurrence of a dropout event.

Naturally, we could get the survival analysis loss function of sample $i$ by taking the negative logarithm of the likelihood function:

$$
\ell^i = (\sum_{t=1}^{T^i} \lambda_t) - y^i \cdot \ln(e^{\sum_{t=1}^{T^i} \lambda_t} - 1)
\tag{11}
$$

Then, the overall loss is the sum of the loss functions of all samples in the training set:

$$
L = \sum_{i=1}^{N} \ell^i = \sum_{i=1}^{N} \left[ (\sum_{t=1}^{T^i} \lambda_t) - y^i \cdot \ln(e^{\sum_{t=1}^{T^i} \lambda_t} - 1) \right]
\tag{12}
$$

By optimizing the model parameters, we can estimate the hazard rate $\lambda_t$ for new samples.

The corresponding survival probability at each timestamp can be calculated by $S_t = e^{-\sum_{k=1}^{t} \lambda_k}$, which is monotonically decreasing along time to helps us achieve consistent predictions. Then we can predict the student dropout result by comparing the survival probability at the last timestamp with an optimal threshold $\tau$.

## Experiment

### Dataset description

We conduct experiments on two SDP benchmark datasets to evaluate our proposed model. They are drawn from XuetangX, the largest MOOC platform in China. It was launched in October 2013 and has provided over 1,000 courses and attracted more than 10,000,000 registered users. The XuetangX platform provides various course information (course start date, course end date, course category, and course type) and user profiles (gender, birth year, and education level). They also record system logs of users' learning activities such as video watching, forum discussion, assignment completion, and web page clicking. Considering data privacy and ethical issues, all the publicly available datasets from the XuetangX platform have anonymized the user names into UserIDs [39].

**Two datasets.**   The first dataset is KDDCup 2015, which has been widely used for various MOOC dropout prediction studies [2]. The dataset contains information about 39 courses and 72,395 enrolled students. Each course takes 30 days as the history time window, and the prediction period is set to 10 days. Seven different event types of student learning activity (i.e., features) are provided in KDDCup 2015. The second dataset is XuetangX [40], which is much larger than KDDCup 2015. The original XuetangX dataset contains 246 courses, 202,000 students, and 22 event types. To facilitate model training, we adopt the data processing method used in [24], i.e., pruning all e-courses with less than 350 student trajectories, which leaves us with 19 courses and 23,839 students. For the XuetangX dataset, the history period is 35 days

**Table 1. Datasets characteristics of KDDCup 2015 and XuetangX.**

| Type | KDDCup 2015 | XuetangX |
|------|-------------|----------|
| *Num of courses* | 39 | 19 |
| *Enrollment students* | 72,395 | 23,839 |
| *Max time window length in days* | 30 | 35 |
| *Num of event types* | 7 | 22 |
| *Avg sparsity* | 90.8% | 90.02% |
| *Class distribution* (0 : 1) | 20.7% : 79.3% | 38.7% : 61.3% |

and the prediction period is still ten days. Both datasets are split by the ratio of 70% training sets, 10% validation sets, and 20% testing sets. Due to both the datasets suffering from the unbalanced class distribution, we utilize downsampling on the train and valid set to achieve balanced subsets. We leave the test set with no processing to ensure the generalization of the prediction results. Table 1 summarises the characteristics of the two datasets.

## Hyper-parameters and baselines

In this section, we first introduce parameter settings in our model. Then we describe several baseline methods.

In the volatility modeling network, we set the hidden units of LSTM and the filters of 1D-CNN as 128 to keep them consistent in channel-wise dimension to facilitate the fusion operation of the update gate. In order to study the influence of students' short-term behavior duration on the results, we set different convolution kernel sizes of 1D-CNN as 2, 3, 4, 5 and 6, respectively. The dropout rate at the LSTM branch is set to 0.1. In the sparsity reduction network, we set the number of units in the TM-LSTM cell as 128. The model is trained by backpropagation via Adam optimizer with a batch size of 16 and a learning rate $10^{-4}$.

Depending on the assumptions made in the model, the baseline methods can be subdivided into semi-parametric and parametric models.

1. Semi-parametric models
   The semi-parametric models are built on the PH assumption [26], which leaves the time component of the hazard function $h_t$ unspecified. The adopted semi-parametric models are listed as follows:

   - Cox Proportional Hazard (CPH, 2016, [6]): a commonly used semi-parametric survival analysis model, which assumes that the risk of failure is a linear combination of the covariates.

   - DeepSurv (2018, [28]): a nonlinear extension of the CPH model, which utilized a three layer fully-connected neural network to fit the nonLinear relationship between features and levels of risk.

   - Conditional Survival Forest(CSF, 2017, [27]): a nonlinear survival method, which was utilized to inherently model high-level interaction terms by employing maximally selected rank statistics to obtain an unbiased split variable selection.

2. Parametric models
   Parametric models are classified into two main categories according to the way parameters are used. One assumes that the survival time of all instances follows a particular theoretical distribution, while the other leaves it entirely up to the model to learn the dependencies between covariates and survival time.

- Gompertz (2017, [29]): a generalization of the exponential distribution characterized by the logarithm of the hazard being linear in $t$. It results in an exponentially increasing hazard rate. This distribution provides a remarkably close fit and is often applied to model highly negatively skewed data in survival analysis.

- SAFE (2019, [8]): a deep survival framework that adopts RNN to capture the information of time-varying covariates and map them into time-varying hazard/survival probability.

## Evaluation metrics

We adopt a commonly used survival analysis evaluation metric, Concordance Index (C-index) [41], to evaluate the models' discrimination ability. It can provide a ranking of the survival time based on the individual hazard rate. Specifically, C-index randomly forms pairwise comparisons of all subjects and calculates the proportion of pairs whose predicted hazard rates are consistent with the actual survival time and censored indicators. The C-index values of survival models generally range from 0.5 to 1, where 0.5 is equivalent to a random prediction result, and 1.0 is a perfect concordance. Additionally, we employ Accuracy, F1, and Area under the ROC Curve (AUC) to evaluate the classification performance of various models. Actually, AUC is the probability that the model ranks a random positive example more highly than a random negative one. According to the definition of C-index, it is a generalization of AUC.

## Results

### Results of different models

We perform five random runs and report the average testing performance of our proposed method compared with other models. Tables 2 and 3 show the overall results of the comparative experiments on KDDCup 2015 and XuetangX dataset.

We can observe that the proposed model SAVSNet outperforms other state-of-the-art methods on the two datasets. This may be attributed to the fact that we take the advantage of neural network to learn the non-linear relations between the hazard rates and the student's dynamic learning sequence, while effectively addressing the volatility and sparsity of the time series. SAFE achieves the second best results on KDD2015 dataset. This could be due to the limitation in the capability of RNN in modeling the volatile and sparse time series. Interestingly, Gompertz performs better than SAFE over some metrics on the XuetangX data. The potential reason may be felt that Gompertz specifies the negatively skewed distribution of

**Table 2. The results on KDDCup 2015 dataset.**

| KDDCup 2015 dataset | | | | | |
|---|---|---|---|---|---|
| **Category** | *Models* | **C-index** ↑ | **Accuracy** ↑ | **F1** ↑ | **AUC** ↑ |
| Semi-parametric models | *CPH* [6] | 0.7827 | 0.7161 | 0.7993 | 0.7233 |
| | *DeepSurv* [28] | 0.7826 | 0.7274 | 0.8120 | 0.7079 |
| | *CSF* [27] | 0.8118 | 0.7668 | 0.8407 | 0.7565 |
| Parametric models | *Gompertz* [29] | 0.8237 | 0.7855 | 0.8576 | 0.7466 |
| | *SAFE* [8] | <u>0.8753</u> | <u>0.8361</u> | <u>0.8929</u> | <u>0.8023</u> |
| | *SAVSNet (Ours)* | **0.9066** | **0.8464** | **0.8999** | **0.8149** |

[†] indicates that the higher the value, the better the performance. Of all the results, the highest are shown in **bold**. The second highest results are shown above <u>underlines</u>.

**Table 3. The results on XuetangX dataset.**

| XuetangX dataset | | | | | |
|---|---|---|---|---|---|
| Category | *Models* | C-index ↑ | Accuracy ↑ | F1 ↑ | AUC ↑ |
| Semi-parametric models | *CPH* [6] | 0.7464 | 0.7286 | 0.8202 | 0.6464 |
| | *DeepSurv* [28] | 0.5927 | 0.6461 | 0.7589 | 0.5604 |
| | *CSF* [27] | 0.7467 | 0.7313 | 0.8172 | 0.6831 |
| Parametric models | *Gompertz* [29] | 0.7039 | 0.7550 | 0.8425 | 0.6477 |
| | *SAFE* [8] | 0.7762 | 0.7523 | 0.8336 | 0.6986 |
| | *SAVSNet* (Ours) | **0.7976** | **0.7794** | **0.8518** | **0.7384** |

↑ indicates that the higher the value, the better the performance. Of all the results, the highest are shown in **bold**. The second highest results are shown above underlines.

time-to-event, which may closely fit the XuetangX dataset but can not exactly match the time distribution of KDD2015 dataset. The experimental results also show that the overall performance of parametric models is better than semi-parametric models. It appears that these semi-parametric models are subject to the PH assumption, which may be attributed to the deficiency of temporal information in modeling survival data. Among all semi-parametric models, CSF performs better than CPH and DeepSurv in most metrics. It may be reasonable to suppose that CSF reduces split variable selection bias by maximally selected rank statistics. DeepSurv gets the similar results with CPH on KDD2015 dataset but performs worse among other models on XuetangX dataset. It can be inferred that DeepSuvr's use of a three-layer fully connected neural network to fit the relationship between covariates and hazard fuction is not always effective.

## Ablation study

In this section, we conduct an ablation study on KDDCup 2015 dataset to investigate the effectiveness of various components in SAVSNet. With the ablation study, we can get deep insights into the contribution of different components to the final performance.

**Effectiveness of key components of SAVSNet.** Firstly, we define some variations of SAVSNet which denote missing/replacing one or more parts of the model. Specifically, SAVSNet-VM and SAVSNet-SM denote we remove the volatility modeling network or the sparsity modeling network from SAVSNet, respectively. SAVSNet-TM indicates the TM-LSTM unit is replaced with a standard LSTM unit. Except for the changes mentioned above, other components remain unchanged. The experiment results are listed in Table 4. We can find that SAVSNet achieves the best performance on the KDD2015 dataset, which proves the effectiveness of each component of our proposed model. Moreover, it is observed that SAVSNet-SM performs worse than SAVSNet-VM. This could be inferred that the sparsity of data has a more significant influence on the results, so the performance degradation of removing the sparsity modeling network is more prominent. A significant drop in SAVSNet-TM is also noted, which is evident that the introduction of informative missing patterns significantly contributes to the model prediction performance.

**Effectiveness of key components of volatility modeling network.** To further evaluate the design of the volatility modeling network, we remain the sparsity modeling network but change the composition of the volatility modeling network. Specifically, SAVSNet-CN and SAVSNet-LM denote we removed the 1D-CNN part and the LSTM part of the volatility modeling network, respectively (the update gate has also been removed). SAVSNet-UG

**Table 4. Ablation study of key components of SAVSNet.**

| Model | C-index ↑ | Accuracy ↑ | F1 ↑ | AUC ↑ |
|---|---|---|---|---|
| *SAVSNet(Ours)* | **0.9066** | **0.8464** | **0.8999** | **0.8149** |
| *SAVSNet*-VM | 0.8636 | 0.8386 | 0.8948 | 0.8022 |
| *SAVSNet*-SM | 0.8052 | 0.8240 | 0.8830 | 0.8069 |
| *SAVSNet*-TM | 0.7849 | 0.8091 | 0.8711 | 0.8053 |

-VM and -SM denote removing the volatility modeling network or the sparsity modeling network from SAVSNet, respectively. -TM indicates replacing the TM-LSTM unit with a standard LSTM unit.

indicates replacing the update gate with a ratio of 0.5:0.5. As shown in Table 5, SAVSNet still gets the best performance. Noticeably, the overall performance of SAVSNet-CN is lower than SAVSNet-LM. It may be reasonable to suppose that 1D-CNN is more prominent because it provides the ability to filter the random peaks in the raw data. Interestingly, when we focus on SAVSNet-UG, we find that its performance is even lower than SAVSNet-CN and SAVSNet-LM on some of the metrics. It appears that simply fusing the output of the 1D-CNN and LSTM may be difficult to learn the dynamic changes in volatility, which implies the update gate has a substantial impact on the prediction performance.

## Sensitive study

To assess the effect of different 1D convolution kernel size on model performance, we conduct sensitive experiments on the KDD2015 dataset. This approach is a common technique used to compare different versions of the same factor and is seen as a critical component of internal reliability [42]. Specifically, we refer to SAVSNet-n as the model with *n* kernel size of 1D CNN. Table 6 shows the performance of the model with different convolution kernel size from 2 to 7. Then, we choose the results on C-index and F1 and plot their curve. As shown in Fig 4, the C-index and F1 of the SAVSNet-n model first increase from SAVSNet-2 to SAVSNet-3 and then decrease as the kernel size gets bigger from SAVSNet-4 to SAVSNet-7, where SAVSNet-3 has the best performance. It indicates that setting the kernel size to 3 is an appropriate choice to filter out the sequence's volatility, which leads to better model performance.

## Case study

One of the advantages of survival models over binary classifiers is their ability to generate monotonically decreasing survival curves. It is helpful to characterize the time-to-event distributions and distinguish the survival differentiation between students.

**Table 5. Ablation study of components of volatility modeling network.**

| Model | C-index ↑ | Accuracy ↑ | F1 ↑ | AUC ↑ |
|---|---|---|---|---|
| *SAVSNet(Ours)* | **0.9066** | **0.8464** | **0.8999** | **0.8149** |
| *SAVSNet*-CN | 0.8648 | 0.8088 | 0.8708 | 0.8061 |
| *SAVSNet*-LM | 0.8959 | 0.8255 | 0.8836 | 0.8135 |
| *SAVSNet*-UG | 0.7519 | 0.8235 | 0.8826 | 0.8074 |

-CN and -LM denote removing the 1D-CNN or the LSTM part from the volatility modeling network, respectively. -UG indicates replacing the update gate with a ratio of 0.5:0.5.

**Table 6. The effect of different convolution kernel on KDDCup 2015.**

| Model | C-index ↑ | Accuracy ↑ | F1 ↑ | AUC ↑ |
|---|---|---|---|---|
| SAVSNet − 2 | 0.9072 | 0.8158 | 0.8762 | 0.8104 |
| SAVSNet − 3 | **0.9129** | **0.8464** | **0.8999** | **0.8149** |
| SAVSNet − 4 | 0.8949 | 0.8319 | 0.8887 | 0.8140 |
| SAVSNet − 5 | 0.9066 | 0.8138 | 0.8745 | 0.8102 |
| SAVSNet − 6 | 0.8875 | 0.8297 | 0.8870 | 0.8138 |
| SAVSNet − 7 | 0.8955 | 0.8358 | 0.8919 | 0.8116 |

To obtain a general view of the predictions, we apply the K-Means algorithm [43] on the test set of the KDD2015 dataset to group similar instances into clusters by making use of the average covariates of students as input features. Due to the number of cluster $k$ being unspecified, we first calculate the silhouette score [44] of cluster $k$ from 2 to 7. As shown in Fig 5A, the silhouette score is highest when $k$ is set to 5. Then, we assign the students in the test set into 5 clusters and plot the clusters in Fig 5B.

Since the analysis of the clustering results is not the focus of this paper, we only select the instance closest to the centroid of each cluster as the representative instance to compare the survival probability curve predicted by SAVSNet. As shown in Fig 5C, there are apparent differences in the survival probabilities of different cases. The instance of cluster 4 has a higher survival probability initially, but the survival curve has a sharp drop very rapidly. It may suggest that the student in cluster 4 may have been interested in the course at first, but his/her interest shifted as the course progressed. The instance of cluster 3 has a dramatic drop in survival curve from the beginning of the course, revealing that students in cluster 3 tend to drop out of the course at an early stage. The student in cluster 1 and 2 provide a survival curve with a relatively constant slope. In contrast, cluster 5, constituting only 7% of the test data, yields a significantly higher survival probability than other clusters. It may represent that only around 7% of them were interested in continuing their studies. Furthermore, we observe that many of the curves seem to have lower survival probabilities at the end of the observation window (drops in the survival curves around 15 days), which could be speculated that these instances have dropout of the course.

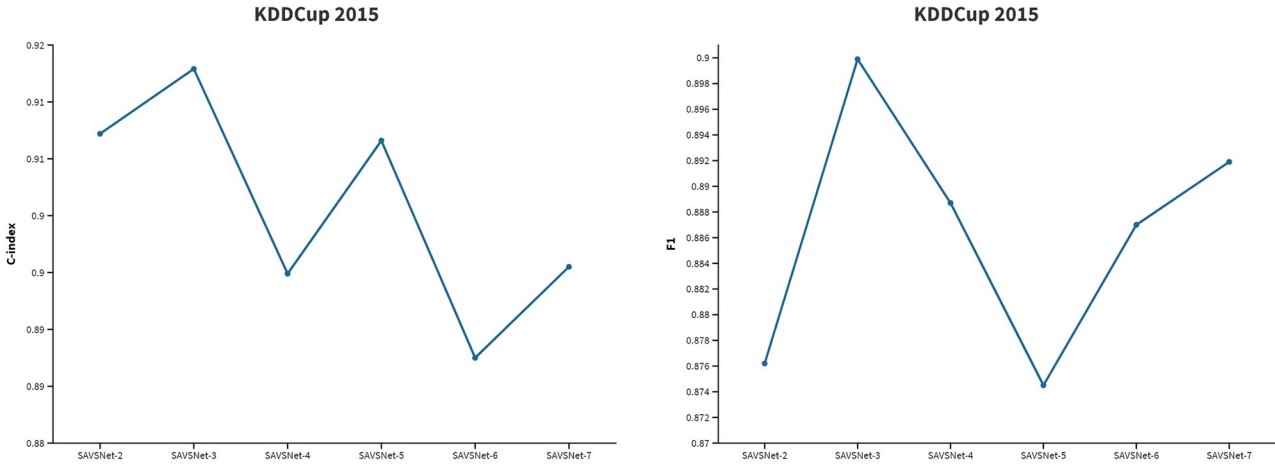

**Fig 4. The results of C-index and F1 on KDDCup 2015 with different convolution kernel size.**

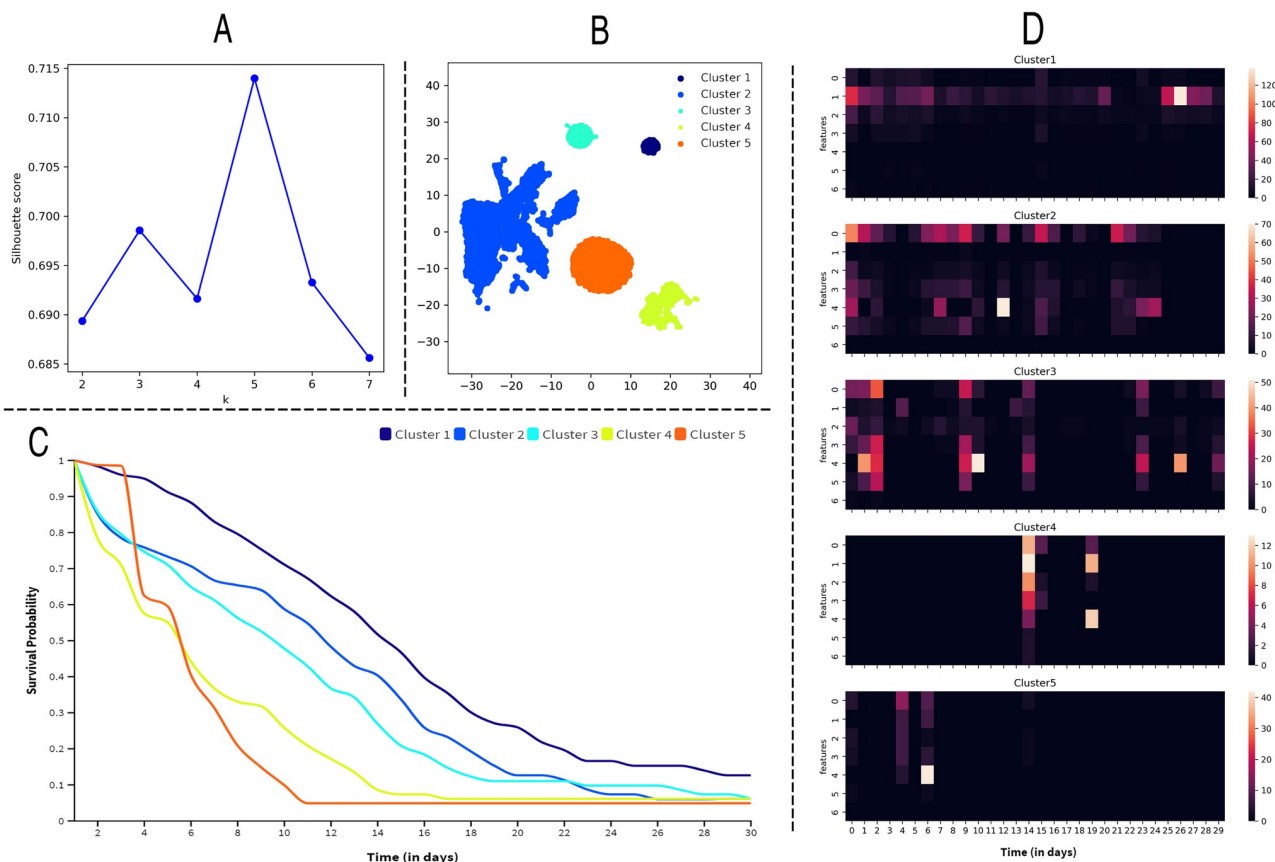

**Fig 5. A case study of SAVSNet.** A: The silhouette scores of clusters from 2 to 7. B: Exhibiting unsupervised clustering into 5 clusters. C: Comparison of the survival probability curves of the representative instance in 5 clusters. D: A set of heatmaps showing longitudinal and multivariate learning activities for the 5 representative instances.

## Discussion

This paper explored a survival analysis model built on neural networks for volatile and sparse sequential data and applied it for the MOOC SDP task. The experiment results suggest that SAVSNet archives a 2% to 3% improvement in C-index and AUC on two real-world datasets compared to the state-of-the-art survival analysis model, which proves that our proposed model is a promising alternative to the naive baseline algorithms [6, 8, 27–29]. Meanwhile, the ablation study results show that removing either the volatility modeling network or the sparsity modeling network will reduce model performance. It demonstrates that our designed modules can effectively deal with the issue of data volatility and sparsity, and the techniques proposed can be generalized to any volatile and sparse sequence scenarios. We also found that the sparsity of the data has a more significant impact on the results than the volatility. The finding is consistent with intuition because sparse data may indicate a lower willingness for students to participate in courses and a higher tendency to drop out. Among all the volatility modeling network components, the elimination of the CNN module results in more significant degradation of model performance. The results validate that activity burst affects the parameter estimates and that the CNN structure effectively filters the volatility over time. Actually, the findings are in line with the results from prior research [33, 34, 45].

Compared with binary classification models, which only output a single prediction outcome [2, 5, 24], our model is a marked improvement in providing a process evaluation for students' learning activities. By mapping the longitudinal students' learning activities into survival probabilities, we can intuitively observe students' performance with the survival probability curve. Therefore, instead of waiting for the end of each course phase to warn students, we can target at-risk students at the early stage of the declining survival curve. It can help us achieve early dropout prediction with fewer data, thus making institutional retention efforts more efficient and effective. Furthermore, an application of event history modeling would assist researchers in examining the temporal evolutionary effects of students' learning activities. By comparing Fig 5C and 5D, it can be found that when students' learning behaviors are sparse, their survival curve will decline rapidly, which indicates a fast transition from persistence to dropout. In addition, we found that when students' activity bursts, their survival curves decline more slowly, indicating that students' efforts can slow the downward trend of the survival curve. It may reveal the causes and consequences of the low engagement of students that frustrate many MOOC providers.

However, we also should acknowledge some limitations, which provide directions for future studies. First, the way the dropout label is defined in the SDP task limits our model's classification performance. As the SDP task generally provide a binary dropout label after the last day of each course [2, 6], we have to compare the survival probability at the last moment with a static threshold obtained from the grid search to convert the survival probability into a binary result. As illustrated by Fig 5C, survival curves with different slopes may fall to zero at the last moment. As a result, it is hard to choose a proper threshold for classifying non-dropouts from dropouts. Further research could select datasets with sequence labels or leverage an automatic threshold mechanism to improve classification performance. Second, the assumption of monotonically decreasing survival probabilities [7, 8, 28] may not be applicable in all situations. For example, a student's commitment to learning could recover the survival probability. Future researchers could design a survival mechanism with a recoverable factor based on this suggestion so that the survival curve is more differentiated at the last moment, facilitating the selection of the optimal threshold.

## Conclusion

In this paper, we propose a novel end-to-end framework named SAVSNet for student dropout prediction, which focuses on addressing the problems of data volatility and sparsity. To reduce data volatility's impact while retaining the original data information, we develop a volatility modeling network to adaptively filter the volatility in multivariate time series by 1D-CNN while retaining temporal correlation with LSTM. Furthermore, to efficiently take advantage of the information of sparse time series, we design a Time-Missing-Aware LSTM unit to incorporate two kinds of informative missing patterns for regulating the flow of information in the LSTM unit. Eventually, SAVSNet provides a process evaluation method by outputting monotonically decreasing survival probabilities. The experiment results on two real-world datasets validate that the treatment of volatility and sparsity improves predictive performance. Extensive ablation studies demonstrate the effectiveness of each component. The broader impact of this study is to indicate that students' engagement and persistence are conducive to reducing the dropout rate. Therefore, MOOC stakeholders should pay close attention to students who have been disengaged for a certain period and provide individually tailored support prior to student withdrawal.

The study is limited by the lack of sequence labeling or an automatic threshold mechanism to achieve better classification performance. These limitations should be addressed in future

studies. In addition, we will design a recovery factor for the survival analysis mechanism to make the survival curve recoverable according to changes in student learning.

## Supporting information

**S1 File. Supporting equations and data.**
(DOCX)

## Acknowledgments

The authors are grateful to Zhang Chunhong for providing insightful feedback on earlier drafts of this paper. We also want to thank anonymous reviewers and editors for providing helpful comments. Additionally, we would like to gratefully acknowledge the organizers of KDD Cup 2015 and XuetangX for making the datasets available.

## Author Contributions

**Conceptualization:** Feng Pan, Yang Ji.

**Data curation:** Bingyao Huang.

**Formal analysis:** Feng Pan, Bingyao Huang, Zhenyu Wu.

**Funding acquisition:** Yang Ji, Zhanfei Ma.

**Investigation:** Zhengchen Li.

**Methodology:** Feng Pan, Zhenyu Wu.

**Project administration:** Zhanfei Ma.

**Resources:** Zhanfei Ma.

**Software:** Feng Pan, Moyu Zhang.

**Supervision:** Yang Ji.

**Validation:** Feng Pan, Moyu Zhang.

**Visualization:** Feng Pan, Moyu Zhang.

**Writing – original draft:** Feng Pan.

**Writing – review & editing:** Feng Pan, Chunhong Zhang, Xinning Zhu.

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
