## [Decision Letter · Decision Letter 0]

24 Jan 2022

PONE-D-21-35650A Survival Analysis based Volatility and Sparsity Modeling Network for Student Dropout PredictionPLOS ONE

Dear Dr. Pan,

Thank you for submitting your manuscript to PLOS ONE. After careful consideration, we feel that it has merit but does not fully meet PLOS ONE’s publication criteria as it currently stands. Therefore, we invite you to submit a revised version of the manuscript that addresses the points raised during the review process.

We look forward to receiving your revised manuscript.

Kind regards,

Zhihan Lv, Ph.D.

Academic Editor

PLOS ONE

Journal Requirements:

2. PLOS requires an ORCID iD for the corresponding author in Editorial Manager on papers submitted after December 6th, 2016. Please ensure that you have an ORCID iD and that it is validated in Editorial Manager. To do this, go to ‘Update my Information’ (in the upper left-hand corner of the main menu), and click on the Fetch/Validate link next to the ORCID field. This will take you to the ORCID site and allow you to create a new iD or authenticate a pre-existing iD in Editorial Manager. Please see the following video for instructions on linking an ORCID iD to your Editorial Manager account: https://www.youtube.com/watch?v=_xcclfuvtxQ.

“This research is supported in part by the National Natural Science Foundation of China (Grant No: 61762071), in part by Key-Area Research and Development Program of Guangdong Province (Grant No: 2020B0101130013) and in part by Baotou Teachers’College High Level Research Incubation Project (Grant No: BSYKJ2021-WY04).”

“This research is supported in part by the National Natural Science Foundation of China (Grant No: 61762071), in part by Key-Area Research and Development Program of Guangdong Province (Grant No: 2020B0101130013) and in part by Baotou Teachers' College High Level Research Incubation Project (Grant No: BSYKJ2021-WY04).”

6. Thank you for stating the following financial disclosure:

 “This research is supported in part by the National Natural Science Foundation of China (Grant No: 61762071), in part by Key-Area Research and Development Program of Guangdong Province (Grant No: 2020B0101130013) and in part by Baotou Teachers' College High Level Research Incubation Project (Grant No: BSYKJ2021-WY04).”

Reviewers' comments:

Reviewer's Responses to Questions

**Comments to the Author**

1. Is the manuscript technically sound, and do the data support the conclusions?

Reviewer #1: Yes

Reviewer #2: Yes

Reviewer #3: Yes

2. Has the statistical analysis been performed appropriately and rigorously? 

Reviewer #1: I Don't Know

Reviewer #2: Yes

Reviewer #3: Yes

3. Have the authors made all data underlying the findings in their manuscript fully available?

Reviewer #1: Yes

Reviewer #2: Yes

Reviewer #3: No

4. Is the manuscript presented in an intelligible fashion and written in standard English?

Reviewer #1: Yes

Reviewer #2: Yes

Reviewer #3: Yes

5. Review Comments to the Author

Reviewer #1: The article is a valuable and original scientific contribution, so I must congratulate the authors for their great work. However, it would be advisable to resolve some shortcomings before being published.

1- Research questions, that drive the paper, should be built in the introduction from an ongoing and pertinent bibliography (up to 2021). These should be of global interest and not focused to a particular local problem. Identifying a research gap is not enough; key is showing its significance to the field.

2- Answer your research question in the conclusions; what did we learn compared with current, significant research .

3- How general are your results? These have to be of interest to the whole community. Relate these with your limitations. .

4- In the conclusion section, the limitations of this study, suggested improvements of this work and future directions should be highlighted.

5- The research problem is not clear, and it needs to be supported and explained why the researchers did the research

Reviewer #2: I am very grateful to the editor for inviting me to participate in the review of this research. I think this research is very interesting, and the authors have done as much analysis and results as possible, which is important. However, I have to point out that this research still needs to be revised, and the authors should ensure that the manuscript can be revised as well as possible before it can be considered for publication.

Title. The authors study the dropout rate of online learning courses, so we should make it clear, otherwise it will be easily confused with traditional dropout rate.

Abstract. Authors should focus their writing on the content of this study instead of spending a lot of time explaining the research reasons.

Introduce. I think the authors should introduce the previous literature analysis in this research field, which can help readers to understand more intuitively why this research is meaningful, rather than the authors' independent explanation. In addition, how is Survival Analysis adopted as an effective method to solve this problem? For a simple example, Survival Analysis has been applied to similar fields and is considered effective. I think the authors can briefly explain it from this angle. The significance of this research is actually more credible than that of self-explanation. In the last paragraph of this part, this is the first time i saw this kind of structure introduction about the full text in the manuscript, and I suggest that the authors adopt experimental design or process to explain it.

Related Works. I don't quite understand this expression adopted by the authors, which is consistent with the suggestions I put forward in my introduction. I think this part is more like a theoretical basis or a literature review. Specific authors can refer to the specific requirements of journals for revising the manuscript framework. At the same time, in this part, I only saw the introduction of three key words, but I didn't see in detail the collation and summary of the authors' previous research on these contents. I think this is very important, because it can highlight the importance and value of this research.

Methods. I think that the authors' calculation of the model is OK, but what needs to be pointed out is that they are relatively weak in literal expression, which makes my reading of the whole calculation process not smooth. I suggest that the authors further sort out the logic of expression.

Experiment. First of all, I need a serious description of this part, which means that it needs to be greatly revised. The authors are not clear about the source of the data. It is not advisable to introduce only some numerical values. Every step of the experiment needs to be explained clearly. At the same time, I realize that if it involves the examination of courses and students, whether this research has obtained the ethical review is very important for publishing, and it is the criterion that authors need to follow. If it is not involved, it needs to be clearly explained.

Result. I actually agree with the authors' statements in the results section, but I still want to suggest whether the authors should incorporate other proven methods to further verify the accuracy of your research, because I personally don't think the existing sensitive analysis is enough to verify the reliability of the research.

Conclusion. Maybe it's because of the space. I don't think the authors have done a good job in this part. I suggest that the authors elaborate on some suggestions of this study for reducing the dropout rate. That is, integrating and analyzing the findings of the existing research, the reality and the findings of this study will help to study the by going up one flight of stairs at the application level, and will also increase the readability of the whole manuscript.

Finally, I appreciate the authors' research very much, but I hope you can make a good revision to ensure that your research can be published. Congratulations on doing a very good research.

Reviewer #3: An interesting study, to investigate the issue of student dropout prediction in massive open online courses with a sound methodological approach. A discussion section should be added after results to elaborate the link of your findings with the prior literature.

6. PLOS authors have the option to publish the peer review history of their article (what does this mean?). If published, this will include your full peer review and any attached files.

Reviewer #1: No

Reviewer #2: **Yes: **Huaruo Chen

Reviewer #3: No

---

## [Author Response · Author response to Decision Letter 0]

9 Mar 2022

We have studied reviewers’ comments carefully and have tried our best to revise our manuscript according to the comments. We hope this revised manuscript has addressed your concerns. The respond to specific reviewer is in the "Response to Reviewers" file.

---

## [Decision Letter · Decision Letter 1]

4 Apr 2022

A Survival Analysis based Volatility and Sparsity Modeling Network for Student Dropout Prediction

PONE-D-21-35650R1

Dear Dr. Pan,

We’re pleased to inform you that your manuscript has been judged scientifically suitable for publication and will be formally accepted for publication once it meets all outstanding technical requirements.

Kind regards,

Sathishkumar V E

Academic Editor

PLOS ONE

Additional Editor Comments (optional):

Reviewers' comments:

Reviewer's Responses to Questions

**Comments to the Author**

1. If the authors have adequately addressed your comments raised in a previous round of review and you feel that this manuscript is now acceptable for publication, you may indicate that here to bypass the “Comments to the Author” section, enter your conflict of interest statement in the “Confidential to Editor” section, and submit your "Accept" recommendation.

Reviewer #1: All comments have been addressed

Reviewer #2: All comments have been addressed

2. Is the manuscript technically sound, and do the data support the conclusions?

Reviewer #1: Partly

Reviewer #2: Yes

3. Has the statistical analysis been performed appropriately and rigorously? 

Reviewer #1: Yes

Reviewer #2: Yes

4. Have the authors made all data underlying the findings in their manuscript fully available?

Reviewer #1: Yes

Reviewer #2: Yes

5. Is the manuscript presented in an intelligible fashion and written in standard English?

Reviewer #1: Yes

Reviewer #2: Yes

6. Review Comments to the Author

Reviewer #1: No Comments

The researchers made appropriate modifications to the article. The article in its current state is accepted for publication.

I thank the efforts of the researchers.

Reviewer #2: I appreciate that the authors were able to make a lot of changes based on the reviewers' comments. However, I don't see that all my previous suggestions have been revised by the authors. Another important point is that the authors should submit vector images, for example in png format when saving. The current images are still blurry and clearly not enough to meet the requirements for publication.

7. PLOS authors have the option to publish the peer review history of their article (what does this mean?). If published, this will include your full peer review and any attached files.

Reviewer #1: **Yes: **usama mohamed ibrahem

Reviewer #2: No

---

## [Editor Report · Acceptance letter]

8 Apr 2022

PONE-D-21-35650R1 

A Survival Analysis based Volatility and Sparsity Modeling Network for Student Dropout Prediction 

Dear Dr. Ji:

I'm pleased to inform you that your manuscript has been deemed suitable for publication in PLOS ONE. Congratulations! Your manuscript is now with our production department. 

Kind regards, 

on behalf of

Dr. Sathishkumar V E 

Academic Editor

PLOS ONE